# *Persistent Suffering*: The Serious Consequences of Sexual Violence against Women and Girls, Their Search for Inner Healing and the Significance of the #MeToo Movement

**DOI:** 10.3390/ijerph18041849

**Published:** 2021-02-14

**Authors:** Sigrun Sigurdardottir, Sigridur Halldorsdottir

**Affiliations:** School of Health Sciences, University of Akureyri, Nordurslod 2, 600 Akureyri, Iceland; sigridur@unak.is

**Keywords:** sexual violence, inter-personal violence, consequences of violence, women’s health and wellbeing, theory synthesis, psychoneuroimmunology, trauma-informed care, victims

## Abstract

This paper uses the method of theory synthesis, primarily from our own previous studies and psychoneuroimmunology research, with the aim of exploring and better understanding the consequences of sexual violence for women and their search for inner healing. The impact of the #MeToo movement is also examined. The main finding is that sexual violence causes persistent suffering for women and girls. In childhood and adolescence, the main consequences include a feeling of unbearable secrecy, threat and humiliation; disconnection of body and soul; great fear and constant insecurity; damaged self-image, self-accusation and guilt; experiencing being compelled to take full responsibility for the crime; as well as various physical and mental health problems, e.g., suicidal thoughts. In adulthood, the consequences are also multifaceted and varied, including vaginal problems, recurrent urinary tract infections, widespread and chronic pain, sleeping problems, chronic back problems, and fibromyalgia, eating disorders, social anxiety, severe depression, and chronic fatigue. In conclusion, sexual violence has these extremely negative and long-term consequences because of the interconnectedness of body, mind, and soul. The seriousness of the consequences makes a trauma-informed approach to services essential to support the healing and improved health and well-being of survivors.

## 1. Introduction

In this paper, knowledge from research on the consequences of sexual violence against girls and women and their search for internal healing is synthesized using the method of theory synthesis. The significance of the #MeToo movement in women’s search for internal healing after sexual violence is also explored and discussed.

When a number of women launched a campaign against sexual violence in 2016, an anti-violence sentiment grew, through which women refused to live in silence about the suffering that many of them had endured because of sexual violence. Women told their stories publicly as part of a movement characterized by support and solidarity. Their suffering had previously been unspoken, an unknown private issue, and this suffering was endured by many as a shameful personal experience.

The #MeToo wave began in the United States within an organization founded by Tarana Burke to raise awareness of the prevalence of sexual violence and to support and empower young women of African descent who had been victims of sexual violence. In October 2017, the #MeToo movement was taken a step further and became integrated via a hashtag on social media. Actress Alyssa Milano was the first to use the #MeToo hashtag and encouraged other women with experiences of sexual harassment and sexual violence to do the same [1]. Since then, the #MeToo movement has reached at least 85 countries and has drawn a great deal of attention to the scale and severity of sexual violence. In the #MeToo revolution, female victims demanded that they be listened to and described systematic gender-based discrimination [2]. 

Women have had difficulty sharing their experiences of sexual violence for fear of the consequences. The movement has shown that sexual violence against women is still a major problem in most cultures and is often linked to differences in power. In a study of 735 girls between the ages of 15 and 19 in the United States, for example, about half of them reported being sexually abused in some way by men who used their position of power against them. According to the study, many girls experienced a great deal of pressure to participate in sexual activities that they were not ready for and 12–28% of the girls reported gross sexual violence [3]. In the wake of the #MeToo movement, women around the world have revolted against sexual violence, and the strength of their voices has led to better access to appropriate resources for survivors and thus opportunities for a better quality of life [4]. 

An international conference on the impact of the #MeToo movement was held in Reykjavík, Iceland, in 2019 and was part of the Icelandic Presidency of the Nordic Council of Ministers. The emphasis was on theoretical and general discussions of the status of women and violence against them due to gender-based inequality, and the violence and harassment around the world that the movement exposed. The conference concluded that it was important to achieve lasting changes in culture, attitudes, and societal values, including the elimination of deep-rooted gender discrimination and the emphasis on zero tolerance towards all forms of sexual harassment and violence.

Why is it important to address sexual violence? What are the consequences of such violence against women and girls? The purpose of this theory synthesis is to examine women and girls’ own experiences of the consequences of sexual violence in childhood, adolescence, and adulthood and what works best for them in their search for internal healing. Therefore, the research questions are as follows: Firstly, what are the consequences of sexual violence for women and girls? Secondly, what is most useful to them in their search for internal healing? In this paper, the authors will also explore and discuss the significance of the #MeToo movement regarding both understanding the consequences and the potential utility of healing.

Before reviewing theory synthesis, the method chosen to answer the research questions, the theoretical background upon which the theory synthesis is based will be explained. In that regard, psychoneuroimmunology research is the most important because it provides an understanding of the close relationship between body, mind, and soul. It therefore gives us an important understanding of the consequences of sexual violence on women and girls. Moreover, it has provided increased knowledge of the vulnerability of individual systems to toxic stress, which occurs in victims of sexual violence, and an increased understanding of the far-reaching consequences of such traumas. In summary, the aim of the paper is to explore and better understand the consequences of sexual violence for women and girls and their search for inner healing.

### Theoretical Background

Our body is designed to be healthy and has built-in balance management to maintain the equilibrium of the various systems of the body. When women and girls are sexually violated, everything in the body tries to prevent the immune system from being damaged and, therefore, decreasing the chances of the person suffering from a physical or mental illness. Psychoneuroimmunology is an interdisciplinary field of study in which the emphasis is on understanding the relationship between what happens to us and the resulting consequences for the central nervous system, endocrine system, and the immune system. Studies in psychoneuroimmunology include an examination of the complex interactions between consciousness, the central nervous system, and the body’s defenses against infections and abnormal cell growth [5].

The body responds to major threats such as sexual violence by secreting catecholamines, norepinephrine and dopamine. This is a biological process that has often been called a “fight or flight response”. This reaction puts the body in a state of reaction and prepares it for conflict. However, it is more appropriate to call it “fight, flight or freeze response”, because a woman or a girl who is sexually violated is much more likely to “freeze” and it can take a long time for her to “thaw” again. The Hypothalamus–Pituitary–Adrenal (HPA) axis responds with reactions involving a chain of events: the hypothalamus releases corticotrophin-releasing hormone (CRH), which causes the pituitary gland to release adrenocorticotropic hormone (ACTH), which causes the adrenal glands to release corticosteroids. Trauma, such as sexual violence, can therefore cause long-term stress (allostatic load), which can activate the HPA axis and the autonomic nervous system. This can manifest as increased or abnormally low levels of cortisol in the blood and low or increased levels of norepinephrine [6]. Psychological stress can trigger the activation of the amygdala and, consequently, the autonomic nervous system, which triggers the immune system and an inflammatory response. Inflammation can increase the risk of psychopathology by altering the metabolism of neurotransmitters. The effect is dose dependent, i.e., the more serious the sexual violence, the more serious the inflammation. Inflammation also increases the risk of cardiovascular disease and type 2 diabetes by affecting the exacerbation of atherosclerosis and the increase in insulin sensitivity [7]. Physical and psychological stress due to sexual violence can therefore cause inflammatory reactions and thus a whole host of diseases.

Excessive and uncontrollable stress has sometimes been called toxic stress, which can reduce the activity of the immune system. It reduces the number of B-cells and T-cells and has the effect of reducing the number and function of natural killer cells. Killer cells can kill harmful cells, such as foreign cells, cancer cells and virus-infected cells. They are, therefore, very important in the body’s defenses. They are activated by interferons that are part of the non-specific immune system.

Human stress responses have several controls and balances built in and designed to prevent hyperactivity. Unfortunately, in the case of overwhelming or toxic stress, such as being sexually assaulted, normal controls and balances sometimes fail, causing the inflammatory response to be abnormally high. For example, cortisol, which is usually anti-inflammatory and retains proinflammatory cytokines, can alter its activity if stress is very high and can increase the activity of interleukin-1 (IL-1) and interleukin-6 (IL-6) rather than inhibiting them. When too many inflammatory cytokines or other inflammatory factors are present, people become more susceptible to disease [8]. Traumatic events increase the levels of inflammatory cytokines in the survivor. Elevated levels of cytokines are associated with an increased risk of serious health problems, including coronary heart disease, myocardial infarction, chronic pain, premature aging, immune response, impaired wound healing, and Alzheimer’s disease, to name only a few [9].

Depression is thought to be a psychological response to overwhelming stress and often overlaps with physical problems, including asthma, rheumatoid arthritis, cardiovascular disease, cancer, and neurological disorders [10]. Inflammation can be the key factor that connects stress and physiological changes [9]. There is growing evidence of a two-way link between major depression and cardiovascular disease [11], and inflammation is thought to be the link between the two [12]. Depression, which may be a psychological response to toxic stress, may make people more susceptible to other diseases because it reduces the activity of killer cells. Severe depression, which accompanies negative events in life, such as sexual violence, suggests that the more depressed someone is, the lower the activity of killer cells is. Furthermore, severe emotional trauma, such as sexual violence, can adversely affect the function of white blood cells by weakening their response to virus-infected cells and cancer cells. Vaccination has been less effective, and wounds heal more slowly in people who have suffered trauma [13]. Toxic stress has also been linked to coronary heart disease, thrombosis, and myocardial infarction due to its effects on the immune system. It can also have a direct effect on the interaction of the nervous system and the immune system and their long-term functioning [14].

Sexual violence is incredibly shocking for victims and the impact of such a shock is immense. The results of many studies have shown a strong link between trauma, stress, and inflammatory processes. The central nervous system, endocrine system and immune system are important systems in the body that communicate abundantly with each other and are key elements in health and well-being. The immune system receives and sends messages, but it is not the endpoint, as previously thought [15]. Research into measurable interactions between these systems enhances our understanding of the psychological link between trauma, such as sexual violence, and disease.

Chronic psychosocial stress caused by sexual violence is an important environmental factor in the development of obesity [16]. Prolonged pressure on the adrenal system in association with chronic stress can disrupt the metabolic rate and cause fat accumulation, hypertension, and diabetes. Obesity and metabolic disorders are more common in people with Post-Traumatic Stress Disorder (PTSD), which is one of the common causes of sexual violence [17]. Stress and depression have also been linked to increased inflammatory responses and risk of disease, including obesity, type 2 diabetes, and cardiovascular disease [18,19,20]. Weight gain can then activate the inflammatory response by increasing the production of IL-6 in enlarged adipose tissue and white blood cells by leptin [21]. The result of research in psychoneuroimmunology is that toxic stress caused by sexual violence can directly lead to obesity.

## 2. Materials and Methods

News coverage of the #MeToo movement has often revealed a great deal of ignorance about the serious long-term consequences sexual violence has on women and girls. In response, the authors decided a summary and theory synthesis of some of their own existing research results about the long-term consequences of sexual violence on women and girls was timely. In addition, since only a portion of the research data is published in each study, there is a large amount of data that researchers have collected and have access to, giving them the opportunity to gain an even deeper understanding of the subject. When researchers do multiple studies on different aspects of the same subject, it is therefore productive to work in more depth with the research data and develop a theory using the research results from these studies. This is the aim of theory synthesis. Walker and Avant [22] believe that theory synthesis is an underused method of theory development and needs to be used much more to further the development of applied science. The main aim of theory synthesis is to formulate a theory from the available research results. This enables theorists to synthesize a considerable amount of data into a single integrated theory. Theory synthesis, in this way, involves three main steps, which are described in Table 1.

### Steps in the Theory Synthesis

Step 1. The bases of the theory are the lived experiences of women who have been victims of sexual violence in childhood, adolescence and/or adulthood and the effects this has had on them, as described in seven of our own published studies [23,24,25,26,27,28,29], as well as their challenging journey towards internal healing. These were used as the bases of the theory in this first step of the theory synthesis. An overview of them can be found in Table 2, as well as what the key factors were in the women’s descriptions of the physical, mental, and social consequences of the sexual violence they experienced, and what benefited them most in their search for internal healing. We extracted key concepts and key statements from the studies and summarized all the factors that helped to answer the research questions.

Step 2. After working from the evidence base that was created in the first step of the theory synthesis, we examined the results of studies and academic writings that had formed a theoretical background in our own studies for more than a decade in the field of sexual violence and psychological trauma. This was done to reach a common conclusion about the consequences of sexual violence for women and girls and what benefits them in the challenging journey of seeking internal healing following such trauma. In this search, we obtained confirmation of our findings in Step 1. First and foremost, we found that psychoneuroimmunology research results on the relationship between body, mind and soul helped us better understand the consequences of sexual violence and women’s challenging journey to seek treatment for their body, mind and soul following such violence and the resulting toxic stress.

Step 3. In this step, the results are presented using the methods that are considered most suitable for the material. We chose to present the theory in the text and in a figure.

## 3. Results

In the presentation of the results, the theory is described in the text and within a figure. The main concepts of the theory are defined in Table 3.

In response to the first research question on the consequences of sexual violence for women and girls, the main findings are that sexual violence causes an enormous psychological shock and results in toxic stress that triggers inflammatory processes that are the bases of many physical and mental illnesses. The consequences are far-reaching, in childhood, adolescence and adulthood. The essence of the long-term consequences of sexual violence is *persistent suffering*.

### 3.1. Consequences of Sexual Violence in Childhood

The consequences of sexual violence in childhood are emotional and existential pain and anguish for the survivors. Following the violence, they are always vigilant, always expecting something bad to happen and feel that their personal defenses have been broken down. They struggle with self-blame, broken self-image, and guilt, leading to great secrecy, intimidation, fear, and humiliation. They also experience multifaceted physical and mental problems and always feel “different” after the violence. They cannot sleep at night and struggle with multiple psychological problems, such as attention deficit disorder (ADD) or attention deficit hyperactivity disorder (ADHD). They experience deep internalized suffering, and the disconnection of body and soul, feel that their defenses have been broken down and experience great vulnerability as a result. Some use sugar as an anesthetic for their emotional and existential suffering and are fighting eating disorders, and often obesity, as a result. Their emotional pain is mostly directed inward, causing *persistent suffering* in the form of inner anguish and despair.

### 3.2. Consequences of Sexual Violence during Adolescence

Following sexual violence in adolescence, the young girls’ lives is difficult and characterized by bullying and distress, a great deal of teasing, few, or no friends, and resulting isolation. They focus their emotions inwardly and are repressed, usually having learning difficulties, struggling with dyslexia and attention deficit. Most try to be invisible and live in constant fear. Most of them struggle with eating disorders, especially obesity, and some start using alcohol during adolescence to soothe their inner pain. They have various physical problems, e.g., myositis, gastritis, abdominal pain, migraine, headache, and gastrointestinal problems. At the same time, they struggle with dizziness and feel faint. Most of them struggle with suicidal ideation and self-harming behavior, and some attempt suicide. In most cases, their emotional and existential pain is directed inward, causing *persistent suffering*, inner anguish, and despair.

### 3.3. Consequences of Sexual Violence in Adulthood

The adult survivors struggle with health problems in the lower abdominal area, such as severe and unexplained pain, recurrent urinary tract infections, ovarian inflammation, irregular menstruation, endometriosis, chlamydia, and ovarian cysts. They also deal with miscarriage, placental abruption, and various other problems that often end in hysterectomy. Other complex and extensive physical health consequences and severe physical symptoms included widespread and chronic pain, high blood pressure, endocrine problems, diabetes, lymphatic problems, abdominal pain, cardiovascular problems, indigestion, neurological problems, asthma, fibromyalgia, myositis, chronic back pain and other musculoskeletal problems. Following the sexual violence, they feel they have lost the joy of life and the will to live. They experience high levels of stress, fear, anxiety, fright, sadness, anger, depression, personality disorder, post-traumatic stress disorder, social phobia, sadness, depression, and postpartum depression. They all struggle with sleeping problems, feel tired and lacking in energy and find it difficult to stand their own ground. They experience heavy flashbacks, difficult memories and nightmares, loneliness, rejection, phobia, and isolation. They experience shame and guilt and feel “dirty”. They try to numb their emotional and existential pain by using food, alcohol, or drugs. They experience great unexplained discomfort, a broken self-image, diminished self-esteem, and self-regard, and have little to no self-confidence, difficulty getting a job, and many are on disability allowance. The survivors also struggle with suicidal thoughts, self-harming ideation, and some attempt suicide. Following the sexual violence, they find it difficult to trust and maintain normal relationships with others, especially men. They also have difficulties regarding touch, sex, and relationships with their spouses. They experience exposure to all kinds of further violence, including rape and repeated physical, mental, and sexual violence. They all experience great anxiety, stress and strain as mothers. Most of survivors often seek help from healthcare services for their various health problems and are usually not asked about experiences of violence. They themselves rarely report their experience of violence and experience little support from healthcare professionals, but a large amount of medication.

The psychoneuroimmunology research findings, which entered the theory synthesis in Step 2, explain that the severity of the consequences of sexual violence is due to the fact that, when a woman or a girl experiences sexual violence, she suffers a severe trauma and the more severe the violence is, the more serious and profound the consequences are. She is likely to “freeze” and not try to escape or fight, which makes sense since the perpetrator is usually bigger and stronger than her. Her involuntary assessment is that she has a greater chance of surviving by freezing. However, this will have long-term mental and physical consequences. Her defenses have been broken down and her sense of self-control is taken from her, which makes her more vulnerable as a result and has very negative consequences.

### 3.4. The Challenging Journey in Search of Inner Healing after Sexual Violence

The women sought internal healing for their invisible psychological traumas. In the aftermath of the sexual violence, they often struggled with physical, emotional, mental, and social problems and tried their best to soothe their severe existential pain by all available means. In response to the significance of the #MeToo movement on the journey of women in the pursuit of internal healing, it is important to first look at the results of a study on the Wellness Program (Study 7 in Table 2), designed to promote the inner healing of women who have experienced sexual violence. It was a 10-week program, organized over 20 hours per week. A group of healthcare professionals used a holistic approach and provided individual and group therapy based on the unity of body, mind, and soul. The women found the most useful treatments to be group therapy, where the emphasis was on disclosing the violence and receiving the support of other survivors, as well as empowerment from supportive professionals. They also felt that they benefited from deep relaxation and hypnosis, body awareness therapy, psychotherapy, group therapy with mindfulness, body therapy with an emphasis on dance and body therapy with massage, as well as craniosacral therapy (CT) provided by a physiotherapist, which is a mild form of treatment aimed at relieving tension in the connective tissue of the body.

### 3.5. The Results in a Nutshell

The results, in a nutshell, are shown in Figure 1. A woman or a girl who is sexually assaulted experiences a psychological trauma. The reaction to such a trauma is to flee or fight, but if that is not possible, the body responds by “freezing”, wherein the overwhelming emotions she experiences lead to “freezing” within her nervous system. Failure to work through these traumatic emotions may result in serious physical and psychological consequences. After experiencing severe trauma, she uses various ways to disconnect from her frozen emotions. Such dissociation can have physical and psychological consequences and can, for example, lead to her developing an addiction and self-destructive behaviour. During the trauma, the woman’s boundaries are disregarded and broken down, making her vulnerable to repeated violence and traumas, as well as addiction. To deal with these consequences, a holistic approach is important, emphasizing the unity of body, mind, and soul. In this approach, the emphasis is on disclosure of the trauma, receiving social support, being able to relax the stress in the nervous system, and self-empowerment. Trauma-informed services are important to prevent re-traumatization. Such services can be implemented in all systems of society, i.e., the school system, the law enforcement system, the healthcare system and the social welfare system and they include, among other things, educating staff about the frequency of trauma and the consequences for the survivor. Subsequently, it is important to offer trauma-informed treatment, that is, treatment offered by specialized therapists who work systematically with trauma survivors to assist them in working their way out of the consequences of the trauma.

## 4. Discussion

Sexual violence is a much more serious crime than most people realize. A woman or a girl who is sexually assaulted, whether in childhood, adolescence, or adulthood, suffers from long-term physical, mental, and social consequences. The most important results are the almost unbearable emotional pain that she usually hides inside herself and her consequent struggles with persistent suffering. This section will address these long-term consequences of sexual violence, the importance of disclosing the violence and the importance of social support for internal healing. Finally, the significance of the #MeToo movement in this regard will be discussed.

What is undeniably striking in the results is how far-reaching the negative consequences of sexual violence are. Why are these consequences so drastic and serious? One of the main reasons for this is that human beings are not a collection of systems, such as the immune system, the endocrine system, and the nervous system, but one whole, and psychological traumas affect all of these systems. A woman or a girl who is sexually assaulted experiences overwhelming or toxic stress that affects the key systems that are part of her stress response. Then, the question is, why does stress have such serious consequences? This is because there is no real difference between mind and body due to the constant communication between the brain, the nervous system, the endocrine glands, and the immune system. Psychological trauma caused by sexual violence has long-term health consequences because such stressful experiences can adversely affect the immune response for the rest of the survivor’s life [30,31,32]. This means that those who have such experiences are at a greater risk of developing serious diseases than those who have not had to endure such trauma. They are more likely to have medically unexplained physical symptoms [33] and have a greater need for healthcare than those without such experiences [34].

Knowledge of the relationship between mind, body and soul and the idea that trauma can cause significant physical and mental changes, as research findings from psychoneuroimmunology show, can provide a framework for examining the healing properties of some methods to promote health and wellbeing after sexual violence. In the future, this could enable healthcare professionals to take a closer look at what works for those who have experienced traumas such as sexual violence. In 2013, a national policy group met in Washington D.C. to review evidence related to the effects of trauma on health and to develop guidelines for healthcare professionals, researchers, and politicians on key aspects of responding effectively to recent and past traumas [35]. The group was established due to numerous studies, growing knowledge and experiences that have shown that people come to health services with common symptoms that can be traced back to traumas. Their goal was to promote trauma-informed care and trauma-informed treatment with the hope of increasing the chances of a cure for those who have been sexually assaulted. Such a group would need to be established in other countries to accelerate the development of trauma-informed services. It is also important to continue researching which treatments are best for girls and women who have been traumatized by sexual violence [36].

It is interesting to note that the women in the Wellness Program felt that what was most helpful for them was when they had the opportunity to open up about the experience of sexual violence, and this is exactly what the #MeToo movement has been all about—disclosure [37]. Research suggests that the reason why disclosing emotional trauma provides the power to heal seems to be related to the fact that inhibition, suppressing emotional pain and distress, requires work. This work consists of physiological exertion that is significantly stressful. Like all other stressors, it raises the levels of catecholamines and other stressors, such as cortisol, in the body, which can contribute to immunosuppression, in addition to damaging the arteries and increasing the risk of various diseases and illnesses. Relieving the suffering caused by violence can only come by disclosing the violence and thus reorganizing one’s thoughts and feelings about the distress created by the violence [38]. Disclosing this inner distress and talking about the violence so that such a reorganization of thoughts and feelings can take place often requires a great deal of courage. What happened in the #MeToo movement is that women saw that they were not alone in this persistent suffering; more women had experienced the same horror. The great solidarity and the powerful message that #MeToo sends to women, “You are not alone”, was what gave women who had been sexually assaulted the strength and courage to disclose the violence they had experienced and return the shame to the perpetrator.

Various studies have been carried out that show the importance of disclosing trauma and not hiding it. For example, 50 healthy college students were asked to write about the psychological trauma they had suffered or a trivial matter, for twenty minutes, four days in a row. Participants who shared much of the traumas they had experienced subsequently showed higher and better mitogenic responses, reduced hyperactivity of the autonomic nervous system, used healthcare less, and their stress was significantly reduced [39]. In another study, the participants consisted of 74 female university students with a history of sexual violence, who wrote either about the sexual violence they had experienced or about how they spent their time during the day. One month later, there was a big change in those who had written about their experiences of sexual violence. They felt very relieved and felt much better in their body, mind, and soul immediately after the writing [40]. Research has also shown that those who systematically disclose difficult thoughts and feelings related to the experience of sexual violence have fewer sick days and doctor visits than people who do not [38]. Meta analyses have been published on the importance of disclosing emotional traumas, such as those caused by sexual violence, which show that such disclosures promote physical health, increase well-being, and enhance the physiological functioning of the body [41].

The #MeToo movement served as a powerful social support for many women who had been sexually assaulted and who had never reported the violence they experienced [1,42]. Social support strengthens the immune system and is one of the most important aspects of healing after sexual violence. It is linked to a lower incidence of the negative consequences that usually result from distress and destruction. In adults, the individual’s psychological characteristics and available social support change the effects of destructive events [43]. Social support acts as a “buffer” and directly protects people from the harmful effects of toxic stress, and those who are active in seeking social support in the aftermath of trauma have an increased quality of life. Social support can therefore reduce the harmful effects of stressful events. It promotes the health of those who experience it, regardless of whether the individual has experienced a stressful event or not. Strong self-help and supportive social networks have a direct effect on the individual’s response to stressful events and reduce the risk of illness following such events. Holt-Lunstad et al. [44] conducted a meta-analysis of social relationships and mortality using 148 studies consisting of over 308,000 participants. Their conclusion is that social support results in a 50% lower mortality rate from all causes and that the relationship between social relationships and deaths is comparable to standard risk factors such as smoking, exercise and obesity. Interestingly, it is the perception of the individual that matters most, which means that experiencing social support through the #MeToo revolution may have done more for women who have experienced sexual violence than many would suspect. It is important that health and social care workers are well informed about the consequences and prevalence of sexual violence, in order to respond correctly to those who seek help, especially when it comes to women who belong to vulnerable groups who traditionally experience prejudice, such as those who deal with mental health problems or excessive alcohol consumption and it is important to focus on trauma-informed services. In this sense, support is key [45].

The #MeToo movement has created an opportunity for healthcare professionals to focus on trauma-informed services for victims of sexual violence through education, targeted action and better resources [46]. Professionals working with victims of violence have also felt the effects of the #MeToo movement. Visits of female victims of sexual crimes to emergency departments, police stations, child protection services and centers for victims of violence have increased. Women seem to seek help sooner than before and are more informed about the resources available. In addition, more women are confessing that they have experienced sexual violence in close relationships than before [47]. However, there have not yet been enough changes to benefit the victims of sexual violence. There still seems to be great responsibility on the part of the victims and the burden of proof is still great. There also remains a lack of research on how best to deal with such issues in the health and social care system. It is therefore important to further strengthen the ways in which, within the healthcare system, changes can be made for the benefit of survivors and to provide trauma-informed services wherever victims seek help [48]. 

## 5. Conclusions

This article focuses on a theory that revolves around the far-reaching physical, mental and social consequences that sexual violence has on women and girls, as well as their search for internal healing and how the #MeToo movement can have a positive effect on that search. #MeToo has had a clear impact on the discussion surrounding sexual violence. However, it is important to further strengthen our education regarding the effects of sexual violence, as well as research in this important area. Sexual violence is a major crime and emphasis must be placed on appropriate responses, considering the need for evidence in favor of the victims. It is important that health and social care professionals consider the fact that one in every three women experiences some form of sexual violence in their lifetime, and that this violence causes persistent suffering for the woman and has a significant negative effect on her health and well-being. It is also important that trauma-informed services are offered wherever women seek health and social services, and that each woman receives the support and treatment she needs to return to a state of improved health and well-being.

## Figures and Tables

**Figure 1 ijerph-18-01849-f001:**
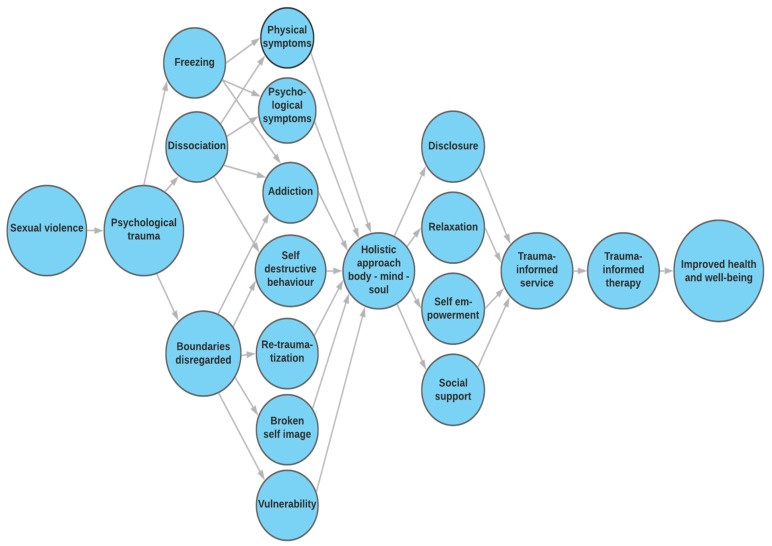
The consequences of sexual violence—trauma-informed approach.

**Table 1 ijerph-18-01849-t001:** Theory synthesis: an overview of the method and how it was used in this study.

Step	Description	Overview of What the Authors Did
*Step 1*	The main concepts and main descriptions specified from the studies used.	We used our own databases and analyses of them as a basis for the theory synthesis. These are studies on the consequences of sexual violence against women and girls, as well as their search for internal healing following such violence (see Table 2).
*Step 2*	In what has been written before, we look for items related to the main concepts or main descriptions and examine the relationship.	We used the table from step one (Table 2) and compared it with research results in peer-reviewed journals where the consequences of sexual violence against women and girls were examined from the perspective of the women themselves. Keywords from the results of Step 1 were used and, with constant comparison, it was possible to examine the consequences of sexual violence against women and girls. Many of the articles were directly related to the consequences of sexual violence for female survivors, while others were only partially related, but could provide important information nonetheless. There was no change due to research in the field of the consequences of sexual violence on women, but research results from psychoneuroimmunology gave more depth to the results.
*Step 3*	The concepts and descriptions that pertain to what the theory is about are systematically grouped together and presented in the text, in table(s) or in figure(s).	After compiling detailed descriptions of the consequences of sexual violence on women and girls, we presented the results in the text and in a figure (Figure 1). The text describes the consequences of sexual violence in childhood, during adolescence and in adulthood. The consequences are physical, mental, and social. The figure describes the consequences of sexual violence on women and girls and how the unity of mind, body and soul must be considered when considering treatment.

**Table 2 ijerph-18-01849-t002:** The seven published studies used in Step 1 of the theory synthesis.

Our Studies	Key Conclusions for the Theory
1. Sigurdardottir, S.; Halldors-dottir, S. Time does not heal all wounds [23].	**Consequences of violence against women:** *Physical symptoms*: multiple health problems in the lower abdomen, widespread and chronic pain, sleeping problems, eating disorders (especially obesity), fibromyalgia, chronic fatigue, cardiovascular problems, and diabetes. *Psychiatric symptoms*: depression (and postpartum depression), anxiety, stress, fear, poor self-esteem, shame, guilt, self-harming behavior, alcohol and drug abuse, severe suicidal ideation; anger, sadness, melancholy, and disappointment; personality disorder, trauma disorder and social phobia. *Social symptoms*: difficulty with touch, sex and relationships with spouses, difficulty trusting men, exposure to all kinds of violence in adulthood, recurring physical, mental and/or sexual violence in a relationship or rape; great anxiety, stress and strain as a parent; seeking health services on a large scale but not reporting the violence—receiving little support but a large amount of medication.
2. Kristinsdottir, A.; Halldors-dottir, S. Constant stress, fear, and anxiety: The experience of women who have experienced intimate partner violence during pregnancy and at other times [24].	**Consequences of violence against women:** Constant stress, fear, fright and anxiety; depression and great distress; increased violence during pregnancy; postpartum depression; broken self-confidence and decreased self-esteem; heavy flashbacks, difficult memories and nightmares; loneliness and isolation; guilt and shame; severe physical symptoms and eating disorders; feeling they cannot stand by themselves; difficulty trusting others; strong feelings of rejection.
3. Sigurdardottir, S.; Halldors-dottir, S. Silent suffering: Long-term consequences of sexual violence in youth for health and well-being of men and women [25].	**Consequences of violence against women:** *In childhood*: emotional pain, agony and anguish; always alert, always expecting something bad and felt their personal defenses had been broken down; self-blame and guilt; great secrecy, intimidation and humiliation; dissociation of body and soul; great fear and constant insecurity; not reporting the violence and receiving even more violence if they try. *Adolescence* was characterized by bullying and great distress; much teasing, few or no friends, isolation; directed their emotions inward and were repressed; usually experienced learning difficulties, dyslexia and attention deficit; most tried to be invisible; lived in constant fear; some started to use alcohol during adolescence to numb their emotional pain; had a variety of physical problems, e.g., myositis and pain, gastritis, migraine, headache, gastrointestinal problems, dizziness and fainting. Some engaged in self-harming behaviors and had suicidal thoughts, and some made suicide attempts. *In adulthood*: many problems in the lower abdomen, unexplained pain; difficulty sleeping, myositis, anxiety and depression, trying to numb the inner pain by using food, alcohol or drugs; self-destructive behavior, self-harming behavior; a strong feeling of rejection; escape, fear and isolation; severe, unexplained malaise; very broken self-image; little to no self-confidence and self-esteem; enormous emotional pain; failing to keep a job; many are on disability allowance.
4. Sigurdardottir, S.; Halldors-dottir, S. Repressed and silent suffering: Consequences of childhood sexual abuse for women’s health and well-being [26].	**Consequences of violence against women:** *In childhood*: They always felt “different” after the violence. Could not sleep at night and struggled with multiple psychological problems. Experienced attention deficit or attention deficit hyperactivity disorder (ADHD). *In adolescence*: A difficult adolescence. Eating disorders. Drinking alcohol very young. Thoughts of suicide and, in some cases, suicide attempts. *In adult years*: Severe pain in the uterine area. Rheumatoid arthritis. Felt they lost the joy of life and the will to live following the violence. Depression. Always tired and lacking in energy. Strong feeling of rejection. Phobia and isolation. Difficulty maintaining normal relationships with others. Marriage problems.
5. Sigurdardottir, S.; Halldors-dottir, S.; Bender, S.S. Consequences of childhood sexual abuse for health and well-being: Gender similarities and differences [27].	**Consequences of violence against women:** *In childhood*: Their emotional pain was directed inward and caused inner agony and despair, as well as deep and silent suffering. They experienced disconnection between their body and soul as well as great secrecy, threat, fear, and humiliation. They were always insecure, felt the need to be constantly vigilant, and always expected something bad to happen. They felt that their defenses had been broken down and experienced great vulnerability. They felt they were held responsible for the violence. *In adolescence*: experienced a broken self-image and a variety of physical and mental problems. *In adulthood*: All have struggled with problems in the lower abdomen, unexplained pain, miscarriage, ectopic pregnancies, multiple inflammation, and problems that often ended in hysterectomy. All of them have struggled with sleeping problems and various physical problems such as fibro- myalgia, high blood pressure, dizziness, endocrine problems, diabetes, lymphatic problems, nervous system problems, asthma, epilepsy and eating disorders (especially obesity).
6. Sigurdardottir, S.; Halldors-dottir, S. Screaming body and silent healthcare providers: A case study with a childhood sexual abuse (CSA) survivor [28].	**Consequences of the violence against one woman:** This was a case study. Since childhood, she has experienced complex and far-reaching physical and health consequences such as recurrent abdominal pain, widespread and chronic pain, sleeping problems, indigestion, chronic back pain, fibromyalgia, musculoskeletal problems, recurrent urinary tract infections, irregular periods, ovarian cysts, ectopic pregnancies, endometrial hyperplasia, inflammation of the ovaries, uterine problems, and ovarian cancer. She told health professionals about her experience of sexual violence to increase their understanding of her health problems, but they remained silent and were unable to provide her with trauma-informed healthcare.
7. Sigurdardottir, S.; Halldors-dottir, S.; Bender, S.S.; Agnarsdottir, G. Personal resurrection: Female childhood sexual abuse survivors’ experience of the Wellness-Program [29].	**Searching for internal healing of the consequences of sexual violence:** The Wellness Program (Gæfusporin) was designed to promote the inner healing of women who had experienced sexual violence. The program lasted for 10 weeks with an organized program of 20 hours per week. A group of healthcare professionals used a holistic approach and provided holistic therapy individually, as well as group therapy, with an emphasis on the unity of body, mind, and soul. In their own view the most useful treatments for the women were as follows: group therapy where the emphasis was on sharing their experiences and gaining the support of other women who had experienced the same, as well as empowerment from supportive professionals; deep relaxation and hypnosis; body therapy with massage; trauma and stress education; craniosacral therapy; psychological group therapy with mindfulness; body therapy with an emphasis on dance; and body awareness therapy.

**Table 3 ijerph-18-01849-t003:** Definition of the main concepts of the theory.

Concepts	Definition
*A woman or a girl who is a victim of sexual violence*	A woman or a girl who has been sexually assaulted is an individual who is part of a family and a community. The violence has made her more sensitive than usual to stress and she needs trauma-informed healthcare and trauma-informed therapy. She is now more vulnerable to various forms of violence.
*Health*	Health has many dimensions, including physical, mental, emotional, social, and societal. A woman’s health can be enhanced or weakened in various ways. A woman’s subjective health consists of her conceptual understanding of her own strengths, which enables her to achieve her most important goals regarding long-term happiness and well-being. Sexual violence has a significant negative impact on every dimension of health of the victim.
*The environment*	A woman’s and a girl’s environment can be divided into two dimensions: the *inner*, which includes the woman’s/girl’s needs, expectations, past experiences and her own self-image; and the *external,* which includes factors outside the woman and the girl that affect her, her family, friends and community. Sexual violence is a very destructive environmental factor that creates a toxic environment for the victim.
*Empowering a woman or a girl who has been sexually assaulted*	A woman’s and a girl’s subjective feelings about being empowered. Self-empowerment reduces a woman’s and a girl’s vulnerability in her situation, increases her well-being, gives her a stronger “voice” in her situation and a greater sense of control of her situation. Empowerment enables her to strengthen herself and cope better with the situation she is in.

## Data Availability

Only published papers were used in the theory synthesis.

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
