# Peer review of "Persistent Suffering*: The Serious Consequences of Sexual Violence against Women and Girls, Their Search for Inner Healing and the Significance of the #MeToo Movement"

_ijerph, 2021, doi:10.3390/ijerph18041849_

Round 1

Reviewer 1 Report

The study does not detail the ethical safeguards that should be considered when studying people who have suffered violence. They must be incorporated.

The work requires greater methodological specifications. Section number 2 must have the following content:
Type of study (qualitative, quantitative or mixed) and justification of the method.

Population and sample
information gathering techniques
Information analysis
The ethical safeguards already mentioned above.

Author Response

 See attahed file 

Reviewer 2 Report

I have reviewed the manuscript entitled “Persistent Suffering: The Serious Consequences of Sexual Violence against Women and Girls, Their Search for Inner Healing and the Significance of the #MeToo Movement”. The main objective of this study was to examine the consequences of sexual violence against girls and women using the method of theory synthesis. Generally, the manuscript makes a very valuable contribution to the field, including current references. However, in my opinion, in some parts of the text, the writing is unscientific. My suggestions are the following:

-Abstract

It could be improved.

I miss a sentence that clearly states what the main objective of the study was.

Line 21: please remove “the authors conclude”, replace with “we conclude” or “as a conclusion”

Keywords do not need to be numbered

-Introduction

The introduction provides sufficient background, but I miss a final paragraph with the main objective of the study.

The first paragraph needs to be revised and improved.

Line 31: remove the first letter “t”

-Materials and Methods.

The methods are clearly described.

-Results

The results are adequately presented.

-Discussion and conclusions are supported by results.

-Line 191. Please check the numbering of the citations. Please check all the citations in text following the journal instructions.
